# Research on Arc Fault Detection Based on Conditional Batch Normalization Convolutional Neural Network with Cost-Sensitive Multi-Feature Extraction

**DOI:** 10.3390/s24237628

**Published:** 2024-11-28

**Authors:** Xin Ning, Tianli Ding, Hongwei Zhu

**Affiliations:** 1State Grid Sichuan Electric Power Research Institute, Chengdu 610041, China; x.ning@stu.xjtu.edu.cn; 2Power Internet of Things Key Laboratory of Sichuan Province, Chengdu 610041, China; 3College of Integrated Circuits, Zhejiang University, Hangzhou 310027, China; zhuhw@zju.edu.cn

**Keywords:** arc fault, CNN, conditional batch normalization, cost-sensitive optimization

## Abstract

An arc fault is a potential hazard in power systems, capable of causing serious safety accidents such as fires. Therefore, the timely detection of arc faults and implementation of circuit-breaking measures are crucial for ensuring safety, preventing fires, and maintaining the stable operation of power systems. Although existing studies have made progress in improving the accuracy of their detection, most methods have not proposed effective solutions that address the cost-sensitive problem of feature selection. Thus, a multi-feature method is proposed by combining time-domain, frequency-domain, energy, and spatial features, which are integrated into a CBN (conditional batch normalization) convolutional neural network for detection. The experimental results show that the proposed method outperforms traditional models in terms of its accuracy and misjudgment rate while maintaining a lower computational cost, demonstrating its superior detection performance. This provides an effective improvement for arc fault detection.

## 1. Introduction

In power systems, arc faults are a dangerous and hidden risk frequently brought on by aging, corrosion, or shoddy wire connections. These arcs can cause fires and equipment failures, leading to several hazardous safety events. Therefore, maintaining the regular and stable operation of power systems, preventing fires, reducing property damage, and ensuring personal safety depend heavily on the quick identification of arc faults and the timely application of circuit-breaking methods. Arc fault circuit interrupters (AFCIs) have been a significant area of study in this discipline in response to the need to recognize and detect arc faults [1]. An AFCI quickly trips the breaker to cut off the power supply when it notices unusual arc occurrences in the circuit, reducing the danger of a fire.

Although arc fault detection products have been developed to a considerable extent, the complexity of AC electrical systems and the high randomness of series arcs make direct modeling quite challenging [2,3]. Thus, the detection and analysis of arc faults require a comprehensive understanding of both time-domain and frequency-domain characteristics to accurately distinguish between normal load behavior and faulty conditions. Time-domain observations reveal that an arc is a sudden occurrence that takes the form of an abrupt transition; normal loads have periodic time-domain transitions, whereas arc faults display chaotic behavior. From a frequency-domain standpoint, the noise spectrum resulting from normal loads usually exhibits stability. However, the noise spectrum resulting from arcs is disordered, especially in the frequency range widely utilized for arc fault detection, which is 1 to 100 kHz. In fault identification, the characteristics obtained from the frequency and time domains are orthogonal, with independent information obtained from each domain. Thus, the efficient use of frequency- and time-domain data for arc fault evaluations may greatly improve detection precision and enable prompt disaster mitigation.

In traditional diagnostic schemes, SVM and Decision Trees are commonly used for classifying signals. The SVM excels in high-dimensional feature spaces, while Decision Trees provide simple, interpretable models, handling complex patterns but potentially struggling with noise in highly dynamic environments [4,5]. At present, many characteristics of feature extraction and arc defect detection are combined, and then neural network algorithms are applied for more sophisticated decision-making. Analyzing the present signals’ variance, mean, and other statistics is a popular strategy among these techniques [6]. The kind and degree of the fault can be further identified by examining how arc faults affect the stability of current signals. Instead of the time-domain aspects described earlier, ref. [7] examines frequency-domain attributes, while ref. [8] looks into arc fault identification from the standpoint of high-frequency energy. Still, most of these studies focus only on increasing detection accuracy, often without additional feature screening that might influence the number of false positives and false negatives seen. While specific characteristics could lower the accuracy overall, they can help reduce false positives. Ref. [9] investigated the use of a dual-channel time-frequency convolutional neural network as an identification algorithm that independently processes time-domain and frequency-domain features. However, because of these features’ independence and haphazard integration into the network, a straightforward concatenation problem compromised its detection performance. Meanwhile, ref. [10] combined information from the time domain, frequency domain, and time–frequency domain to increase detection accuracy by fusing several characteristics. However, there is still an opportunity for improvement in processing and feature selection, particularly in achieving a balance between false positives and false negatives. More careful feature extraction and selection are needed to improve the overall performance of arc defect detection.

It is clear from the studies mentioned above that a wide range of feature extraction techniques have been developed for arc fault detection. However, the problem of cost sensitivity has been primarily disregarded in most studies. Due to multiple judgment conditions and the arc’s continuity, sporadic missed detections and inaccurate arc fault detection have a minimal effect on the system’s overall performance. On the other hand, false positives have the potential to cause circuit breakers to trip the circuit, leading to far more considerable financial losses than those brought on by missed detections. Changing or adding features through feature selection may increase overall accuracy but decrease false positives.

Given that false positives require incredibly high precision, their occurrence should be measured by the number of instances rather than by the proportion of instance. This paper addresses the shortcomings in feature selection found in contemporary research by extracting features from time-domain, frequency-domain, energy, and spatial perspectives, with a primary focus on the spatial extraction of current waveform features, while evaluating the impact of the other three on the performance of the final arc fault recognition system. Furthermore, to address the issue of insufficient integration between extracted features and convolutional neural networks (CNNs), this paper proposes a multi-feature integrated CBN-CNN. This approach enables time-domain, frequency-domain, and energy features to participate in the convolutional abstraction process, allowing for the earlier localization of arc features, thereby improving the performance of arc fault detection.

## 2. Arc Fault Dataset and Detection Methods

### 2.1. Dataset Division and Introduction

Current data from a range of loads in the lab, including standard operating circumstances, series arc fault conditions, and parallel arc fault conditions, were gathered to create our dataset. An arc generator is an experimental tool used to confirm the properties of actual arcs and mimic arc faults. The generator makes it possible to study circuit arcing occurrences. Figure 1 below depicts the arc generator’s construction. ADC sampling was used to obtain the current waveform data after the arc generator was connected to the circuit, either in series or parallel. The dataset is generated using an arc generator, which simulates the arc conditions of various scenarios. This approach effectively suppresses the impact of noise by providing robust training data that include diverse and realistic arc behaviors.

It is worth noting that arc faults cause disturbances in the current data, and fluctuations between different half-cycles are one of the key factors used in identifying fault samples. Considering the impact of historical data and enhancing algorithm correctness, a sliding window approach is used to integrate the prior half-cycle’s data into the current data. The number of arcs within the 0.16 s, which comprises 16 half-cycle data points, determines the sample labels in the produced dataset. We set the threshold for arc occurrence at 4, meaning that the sample is classified as an arc fault if the number of arcs exceeds this value. All arc fault samples with a label of 1 are regarded as positive samples, while arc-free samples are regarded as negative samples and have a label of 0. This strategy enhances noise tolerance by integrating information over multiple cycles, reducing the impact of short-term fluctuations and external interference. Table 1 displays the gathered current dataset, and Figure 2 shows relevant samples.

### 2.2. Multi-Feature Extraction and Cost-Sensitive Selection

It is becoming harder to distinguish between arcs and non-arcs as the working characteristics of AC settings grow more complicated. Current signals are usually used to identify arc faults since their position cannot be predicted [11]. This paper analyzes the characteristics of arcs and non-arcs from four aspects—time-domain, frequency-domain, energy, and spatial characteristics—followed by an in-depth study of arc features to highlight the distinctions between arcs and non-arcs. Figure 3 illustrates our arc fault detection system. It processes arc fault signals through time-, frequency-, and energy-domain feature extraction, while generating an image matrix for the spatial domain. A convolutional network with CBN is used for arc fault diagnosis, classifying outputs as arcs or normal.

#### 2.2.1. Features in the Time Domain

Four features—the current’s maximum, minimum, mean, and variance—are chosen from the gathered current signals. The relevance and usefulness of these features in characterizing the signal’s time-domain properties led to their selection. The maximum and minimum current values stay comparatively constant when the load operates normally. Arc faults, however, result in discernible variations in these two values during an arc fault occurrence [12].

Additionally, the mean value is comparatively steady and less vulnerable to noise interference, representing the overall level of the current. It can be used to evaluate how arc faults affect the safety and stability of the system. The current’s variance describes the degree of current dispersion. The variance rises when an arc fault happens, and the nature and severity of the arc fault can be ascertained by examining this variance.

#### 2.2.2. Features in Frequency Domain

Applying a Fast Fourier Transform (FFT) to the collected signal converts the time-domain signal into the frequency domain to reveal its frequency components. An arc fault’s spectral energy distribution fluctuates significantly because of its instability and nonlinear properties. We use power spectral entropy to examine the spectral energy further [13]. Power spectral entropy quantifies the complexity of a signal’s spectral distribution. Arc faults can be recognized and categorized by comparing the power spectral entropy between normal and arc fault states.

The power spectral entropy (PSE) is derived from the power spectral density (PSD), which is calculated using Equations (1)–(3). The PSD is typically obtained by applying a Fast Fourier Transform (FFT) to the signal to acquire its frequency spectrum. Here, *X*(*ω_i_*) represents the Fourier transform of the signal at frequency *ω_i_* and *N* denotes the length of the signal.

As the load’s operational energy occupies the majority of its spectral energy, reducing the influence of this load-specific energy on arc detection is essential. To address this, the *PSD* of the dominant frequency point, associated with regular load operation, is removed to enhance the accuracy of the detection of arc faults.
(1)PSD(ωi)=|X(ωi)|2N,
(2)p(ωi)=PSD(ωi)∑j=0N−1PSD(ωj),
(3)Entropy_PSD=−∑i=0N−1p(ωi)log2p(ωi).

#### 2.2.3. Features in the Energy Domain

During an arc event, a significant amount of energy is typically generated in the high-frequency regions of the signal, distinctly differing from the conditions observed under regular load operation. Wavelet energy entropy is added to better represent these energy properties [14,15,16]. When an arc fault occurs, the frequency components across various frequency bands decomposed by a wavelet packet analysis change markedly, with the distribution of energy becoming more random and the wavelet energy’s entropy increasing. Moreover, these features help distinguish noise patterns from actual arc fault signals, improving the model’s precision in noisy environments.

Since wavelet energy entropy fluctuates in response to signal changes and varies with harmonic content and frequency components, it effectively characterizes the variations in current signals before and after arc faults. This calculation turns the energy matrix into a probability distribution sequence, where *r*[*i*] represents the probability value of the *i*’th coefficient. The total energy *E_total* of all wavelet coefficients is the sum of individual coefficient energies, denoted as *coeffs*. The probability distribution is given by Equation (4):(4)r[i]=|coeffs[i]|2E_total,
where
$$E\_total=\sum_{i=0}^{N-1}|coeffs[i]{|}^{2}$$

The entropy value of this probability distribution sequence, which yields the wavelet energy entropy, is calculated as shown in Equation (5).
(5)Entropy_energy=−∑i=0N−1(r[i])log2(r[i])

#### 2.2.4. Features in the Spatial Domain

A current signal is a time series signal which lacks spatial characteristics. However, the current signal often exhibits strong periodicity at times when the load usually operates. Multiple periods of current signals can be arranged into a two-dimensional matrix, with each row representing a single sampling period of current values and each column corresponding to the same sampling point across different periods. This 2D matrix format allows convolutional neural networks (CNNs) to extract temporal and spatial features from the current signal automatically.

#### 2.2.5. Feature Selection

After extracting features in the time, frequency, energy, and spatial domains, these features are concatenated and fed into a multi-layer neural network. However, adding more features does not necessarily improve the neural network’s recognition accuracy. Excessive feature input can lead to overfitting and waste computational resources. Therefore, further analysis of these four types of features is required.

In terms of computational complexity, the neural network (typically a two-layer neural network) derives its highest computational cost from its high parallel operation complexity. At the same time, other features are more superficial and require less processing time, as shown in Figure 4. Removing certain features can reduce the probability of missed detections to some extent. However, arc fault detection is a cost-sensitive issue, and minimizing false positives is prioritized over reducing missed detections, as excessive false positives impact actual production outcomes.

In the arc fault detection scenario studied in this paper, the costs of incorrectly judging the presence or absence of arcs are not equal. In this evaluation, we used the number of false positives and the false negative rate as metrics instead of the more common precision and recall. The number of false positives (FPs) is the count of instances where no arc data are incorrectly classified as having an arc. The false negative rate (FNR) is the proportion of arc data misclassified as arc-free data relative to the total arc data.

By retaining spatial features as the primary input and keeping the pre-trained parameters of the two-layer convolutional network unchanged, we selectively removed time-domain, frequency-domain, and energy features from the output layer to observe the final detection results, as illustrated in Figure 5. When retaining these three feature types, there was a slight increase in missed detections, but the false positive rate was significantly optimized, achieving the best performance. Additionally, the added features did not substantially impact computational efficiency because their extraction processes are relatively simple compared to the forward propagation of the neural network, which means we only need to ensure that the overall processing time is within the constraints. In this experiment, the average computation time per sample was 0.7 ms, meeting the demands of real-world arc fault detection scenarios.

### 2.3. CBN Convolutional Neural Network

Traditional batch normalization (BatchNorm) layers are added after convolutional layers to normalize outputs across network layers, accelerating convergence and preventing overfitting. The scale and shift parameters of BatchNorm are learned through backpropagation. However, in conditional batch normalization (CBN), these parameters are generated not directly through learning but through a multi-layer perceptron based on the input features [17].

CBN was first applied as an improvement strategy in visual question answering (VQA) systems, which take both an image and a sentence as inputs. In VQA, each input type undergoes independent feature extraction, e.g., convolutional layers like ResNet are used for images and sequential models for sentences, with the fusion occurring only after feature extraction, limiting integration at lower-level geometric feature extraction. CBN addresses this by allowing the sequential fusion of semantics and allowing image data to be processed early. It modifies specific image features based on natural language inputs, creating a closer link between image abstractions and semantic features, which improves its training speed. This approach to incorporating external feature vectors has since been widely adopted in the GAN field, as seen in conditional GANs (cGANs) [18]. The CBN calculation is shown in Equation (6):(6)CBN(x)=γ⊙x−μ^Bσ^B+β
where **x** represents a batch of input samples, μ^B and σ^B stand for the batch sample mean and variance, and γ and β are the scale and shift parameters learned through MLPs.

Based on this framework, we introduce the CBN structure into the arc fault detection process. The arc features can be viewed as highly abstract waveform descriptions, where CBN layers regulate the convolution process to emphasize arc randomness and current non-periodicity. With minimal computational overhead, the CBN layer enables the convolutional process to incorporate valuable feature parameters, supporting the extraction of current waveform information, accelerating network convergence, and enhancing generalization through guided human experience.

Because features retrieved from distinct segments may have different means and variances, using the extracted features as direct inputs can be risky. This disparity may considerably reduce the efficacy of some features in the data distribution. Before determining the scale parameters and shift parameter for CBN within the multi-layer perceptron, our network carries out normalization to lessen the effects of this uneven data distribution.

Based on the advantages of CBN and convolutional neural networks, we propose a convolutional network structure that integrates multiple feature extractions with CBN. The overall architecture of the network is illustrated in Figure 6. The input to the convolutional network is the arc current waveform. We utilize the extracted time-domain, frequency-domain, and energy features to simplify the neural network’s parameters and incorporate experiential knowledge. These features are processed through a mapping network to compute the scale and shift parameters fed into the CBN layer. This process combines the parameters with the spatial features extracted by the convolutional kernels.

In the later stages of spatial feature extraction, the outputs from the convolutional layers are directly concatenated with the feature inputs to form a fully connected layer. A multi-layer perceptron is then employed to determine the presence of an arc fault. Finally, multiple fully connected layers, along with a Softmax function, classify the output into two categories, “arc present” or “arc free”, completing the algorithmic flow for the arc fault identification device.

By incorporating the CBN layer, which allows for a broader consideration of global sequence relationships compared to the local perception mechanisms of convolutional layers, the network effectively addresses the relationships between different spatial channels. This results in a comprehensive extraction of arc fault data. The neural network’s architecture is detailed in Table 2.

## 3. Experimental Results and Discussion

### 3.1. Training and Prediction Results

During the training process, weights were assigned to the classifications in the loss function to reduce the occurrence of false positives, with the weight for false positives set to 20. The results of the training process are shown in Figure 7. In the initial stages of training, due to measures such as weight decay, used to mitigate overfitting, the accuracy of the test set exhibited significant fluctuations. However, its upward trend remained consistent with that of the training set. In the later stages of training, the accuracy of the training set showed minimal variation, and the test set’s accuracy gradually stabilized, oscillating within a narrow range around 99.7%. This performance surpassed the training set’s, indicating that the network did not experience overfitting. The introduction of CBN allowed the neural network to leverage the strengths of feature extraction and CBN itself, effectively maintaining high accuracy while preventing overfitting and demonstrating a good generalization capability.

### 3.2. Analysis of Model Comparison Results

To better compare the effectiveness of the models and demonstrate the advantages of the CBN network, we trained separate networks for the BN model and a model without feature extraction. Specifically, we implemented two distinct models: FEATURE + BN + CNN and BN + CNN. These were compared against the CBN network, as illustrated in Figure 8. All three models employed the same parameters and training schemes.

After completing the training of the three models, their performance differences on the test set were summarized, as shown in the table below. The table includes Multiply–Accumulate Operations (MACs), which measure the complexity of the algorithms and models.

Table 3 shows that the third network model outperforms the second model in terms of its accuracy and false negative rate on the test set, even though the two models show similar numbers of false positives. The MACs metric, which represents the time consumed by the model during forward propagation, indicates that the CBN model has a lower number of MACs due to the absence of learnable parameters during training, eliminating the need for backpropagation calculations.

In practical applications, preprocessing time is also a critical factor. Although the first model benefits from not requiring preprocessing, its performance is comparatively inferior. Therefore, in scenarios where performance is a priority, the convolutional neural network that integrates multi-feature extraction with CBN provides the advantages of both a simpler architecture and higher accuracy. The result above suggests that adopting the CBN approach enhances the model’s efficiency and effectiveness in arc fault detection tasks, making it a preferable choice for relevant applications.

To further illustrate the effectiveness of our network, we compared the test results with those from other studies, as shown in Table 4. The parameters of the sample dataset can influence the accuracy of different recognition methods. However, the method proposed in this paper achieves a high recognition accuracy even when conditions include a large dataset, diverse load types, and the inclusion of both series and parallel arc faults. The comparison highlights how our model performs relative to established methodologies, reinforcing the claim that integrating multi-feature extraction with CBN significantly enhances our arc fault detection capabilities across varying conditions.

**Table 4 sensors-24-07628-t004:** Test accuracy of the proposed method and other relevant approaches.

Parameter		Method		
		Our Method	Deep Residual Shrinkage Network with Attention [19]	RIME-Res-1DCNN [20]
Arc fault dataset	Sample frequency	6.4 kHz	25 KHz	—
	Dataset size	500 K	4240	14,485
	Preprocessing method	Multi-feature cost-sensitive selection	CWT + PCA	High-frequency feature extraction
	Data dimensions	1 × 1 K	1 × 2.5 K	9
Model performance	Accuracy (%)	99.78	98.52	99.42
	Number of FPs	2	8	16
	FNR (%)	0.35	0.10	0.22

## 4. Conclusions

This paper investigated multi-feature extraction methods used for arc fault detection and proposed a novel detection model based on a CBN convolutional neural network. The model leverages a simple network structure, effectively reducing its complexity while focusing on the integration of multiple features. The key innovation of this paper is the introduction of the CBN structure, which has rarely been used in the field of arc fault detection. By integrating multi-domain features—time, frequency, energy, and spatial—our approach addresses the critical issue of false positives in arc fault detection. The use of conditional batch normalization enhances feature integration and emphasizes arc-specific characteristics. Our experimental results demonstrate that the CBN-CNN model outperforms traditional models in terms of its accuracy and false negative rate on the test set. Additionally, due to the reduction in its learnable parameters during training, the CBN model reduces the computation of MACs, resulting in higher efficiency and lower computational costs in practical applications. Our method maintains a high recognition accuracy even under complex conditions, validating the effectiveness and superiority of the proposed model.

Based on these experimental results, several avenues for further research and development remain. Future studies could explore the inclusion of additional feature types, such as statistical features or other spectrum analysis methods. These features may encapsulate richer information regarding arc fault behavior, potentially improving the model’s recognition performance. Moreover, the challenge of open-set recognition is prevalent in arc fault detection, where arcs from unknown load types may not be included in the training data. Existing models might misclassify these scenarios. Future research could attempt to integrate open-set recognition algorithms or anomaly detection methods to enhance the model’s ability to handle unknown situations effectively. This could significantly improve the robustness and applicability of arc fault detection systems in real-world environments.

## Figures and Tables

**Figure 1 sensors-24-07628-f001:**
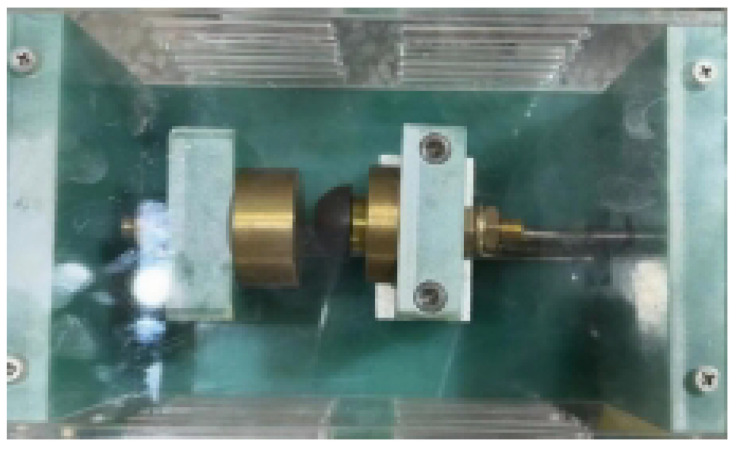
Arc fault generator.

**Figure 2 sensors-24-07628-f002:**
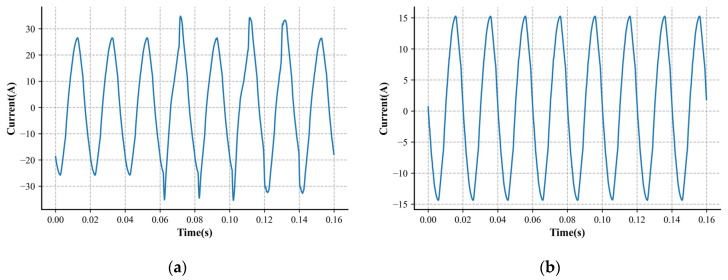
Current of an arc sample and a normal sample. (**a**) Current of an arcing sample. (**b**) Current of a normal sample.

**Figure 3 sensors-24-07628-f003:**
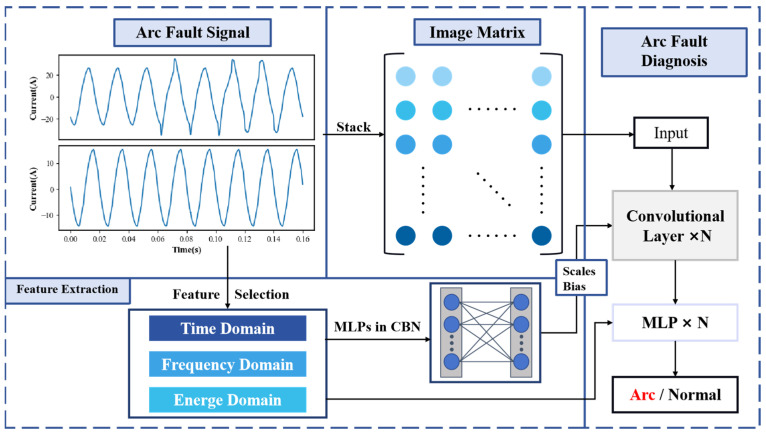
Arc fault detection system flowchart.

**Figure 4 sensors-24-07628-f004:**
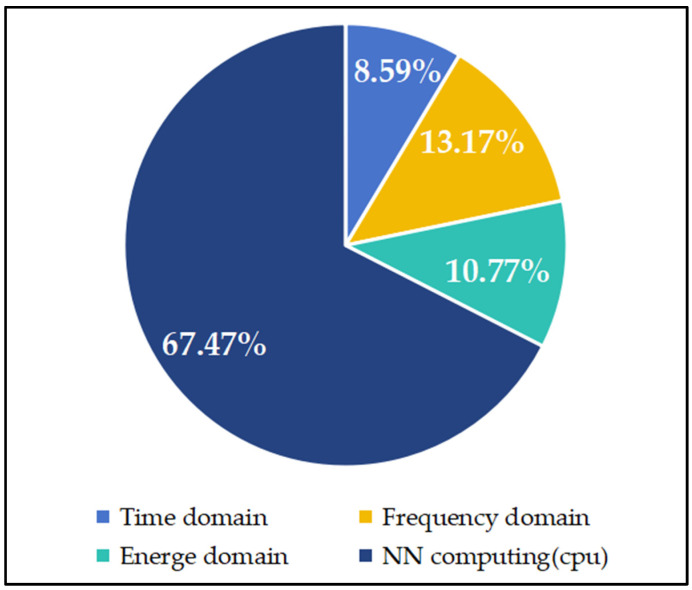
Pie chart of percentage of calculation time attributed to each feature.

**Figure 5 sensors-24-07628-f005:**
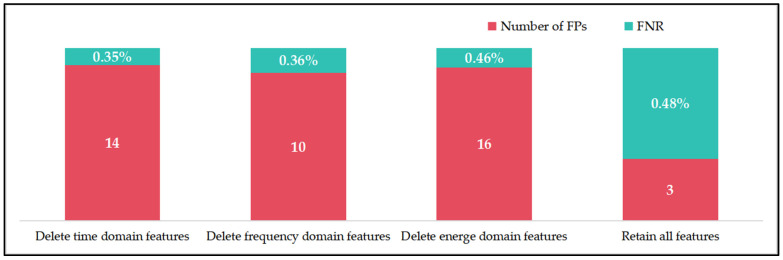
FP and FNR statistics after deleting one feature type.

**Figure 6 sensors-24-07628-f006:**
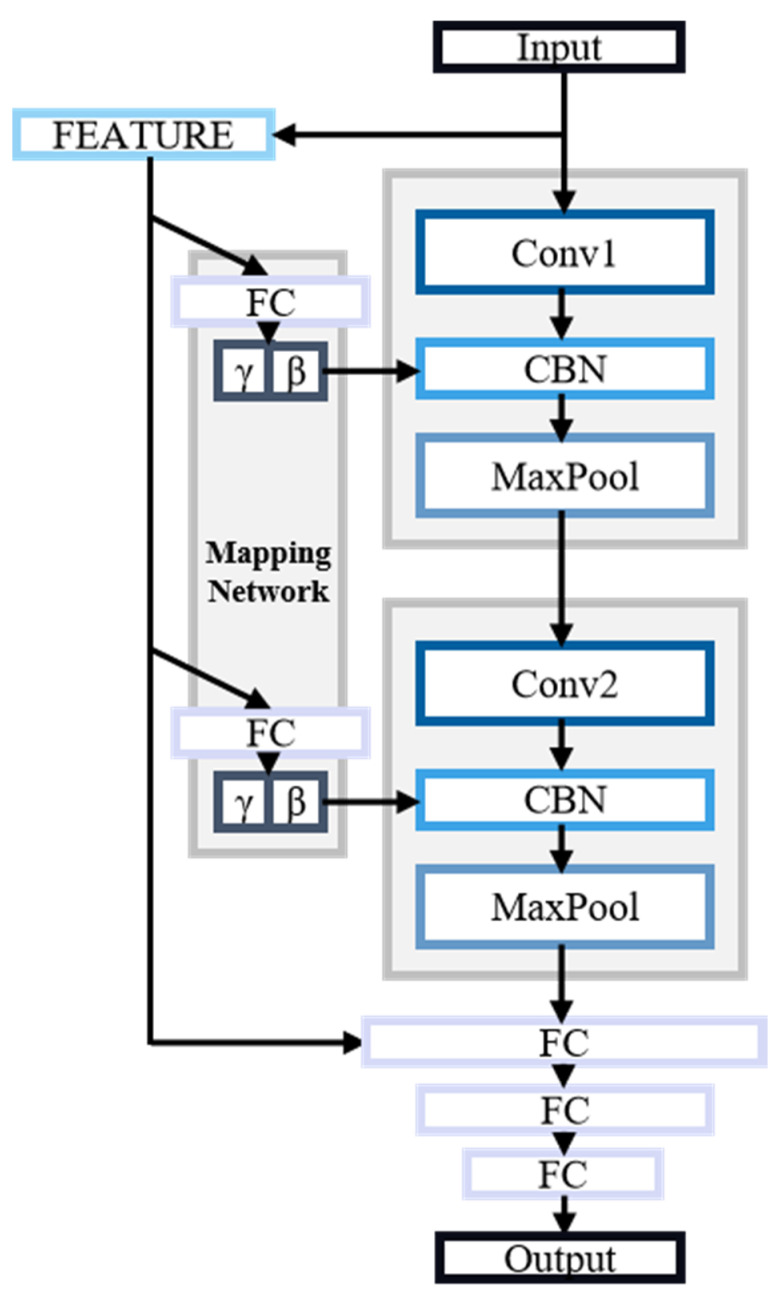
Schematic diagram of the CBN-CNN’s structure. Spatial features are extracted by two convolutional layers. Other features are calculated through the mapping network to obtain the corresponding scaling parameters γ and bias parameters β as the CBN’s input.

**Figure 7 sensors-24-07628-f007:**
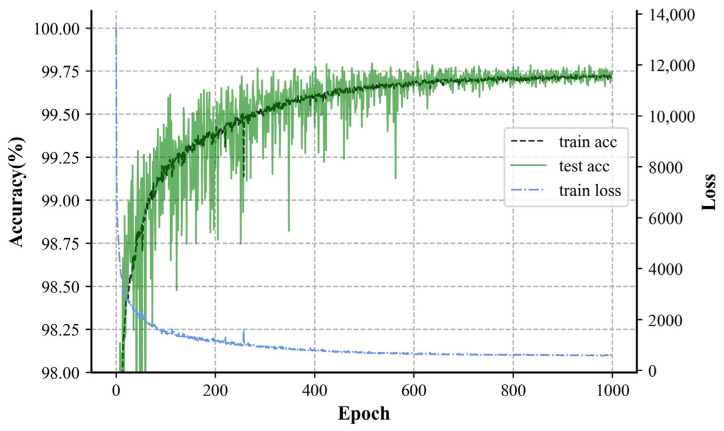
Plot of accuracy and loss changes during training.

**Figure 8 sensors-24-07628-f008:**
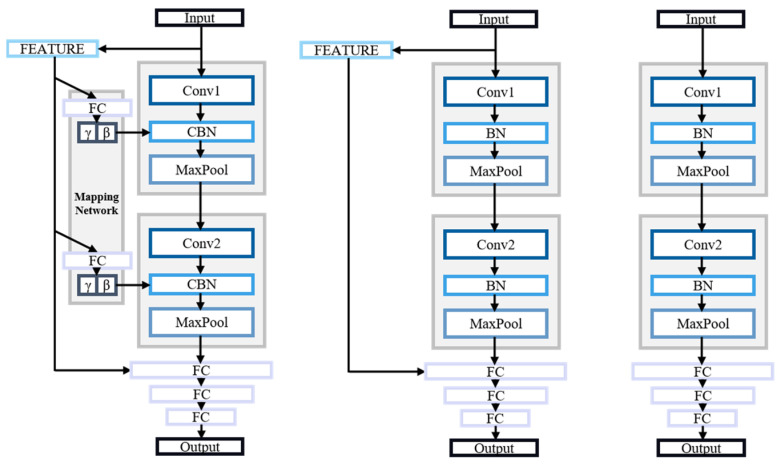
Comparison of network structure diagrams.

**Table 1 sensors-24-07628-t001:** Dataset division result.

Dataset	Normal Samples (Negative)	Arc Samples (Positive)	Total
Training set	769,654	236,822	391,458
Test set	51,892	79,887	131,779

**Table 2 sensors-24-07628-t002:** Parameters of the CBN-CNN.

Path	Layer Name	Output Size	Kernel Size
CNN	Input	1 × 16 × 64	—
	Conv1	4 × 12 × 60	4 × 7 × 7
	CBN1	4 × 12 × 60	—
	Max pool	4 × 4 × 20	3 × 3
	Conv2	8 × 2 × 18	8 × 3 × 3
	CBN	8 × 2 × 18	—
	Max pool	8 × 1 × 9	2 × 2
	Flatten	72	
Mapping Network-CBN1	Input	10	—
	BN + MLP	4 × 2	—
Mapping Network-CBN1	Input	10	—
	BN + MLP	8 × 2	—
Path concatenation	Input	72 + 10	—
	Hidden layer 1	64	—
	Hidden layer 2	16	—
	Output	2	—

**Table 3 sensors-24-07628-t003:** Network identification results.

Model	Dataset	Accuracy (%)	Number of FPs	FNR (%)	MACs (M)
BN + CNN	Training set	99.56	5	1.09	339648
	Test set	99.54	7	1.14	
FEATURE + BN + CNN	Training set	99.67	0	0.46	340928
	Test set	99.68	3	0.54	
FEATURE + CBN + CNN	Training set	99.81	0	0.30	316384
	Test set	99.78	2	0.35	

## Data Availability

The raw data used to support the findings of this study are available from the corresponding author upon request. The processed dataset and code are openly available on Kaggle at https://www.kaggle.com/datasets/tianliding2224/cbn-cnn-dataset (accessed on 9 November 2024).

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
