# Peer review of "Research on Arc Fault Detection Based on Conditional Batch Normalization Convolutional Neural Network with Cost-Sensitive Multi-Feature Extraction"

_sensors, 2024, doi:10.3390/s24237628_

Round 1
Reviewer 1 Report
Comments and Suggestions for Authors
Here are specific descriptions:
1. The abstract on page 1 describes the extraction of time-domain, frequency-domain, and energy features in this paper. However, the spatial perspectives extraction is mentioned in several places, such as the penultimate paragraph on page 2. And it is also emphasized in the penultimate paragraph on page 2 to “with a primary focus on spatial extraction of current waveform features”.
2. There are just only two references before 2024. So the background of this research should be more complete.
3. Page 5, last paragraph, “Additionally, the added features didn't substantially impact computational efficiency”. Please explain why.
4 In Fig.5, the parameters in the structure are not clear. I could not find γ and β in description. And I cannot understand the mean of the gray boxes.
5 The author should explain the novelty and contribution in the arc fault detection. However, the structure and elements in the deep neural network of this paper are regular, and can be found in many papers.
Reviewer 2 Report
Comments and Suggestions for Authors
The manuscript proposes a timely fault arc detection method using CBN-CNN with cost-sensitive multi-feature extraction, aimed at ensuring safety, preventing fires, and maintaining the stable operation of power systems.
1:
The proposed algorithm should include a diagram that clearly illustrates its core concept and key innovations.
2:
How to control the precision and efficiency of fault arc detection, particularly in the presence of external interference and noise signals?
3:
Please highlight several aspects of the usefulness and novelty of this research in the conclusion section. This will greatly help readers better understand the theoretical value and practical significance of this study.
4:
The reference section should be expanded to include more than 20 published papers.
5:
Please provide a detailed performance comparison between the proposed method and relevant approaches. This will help further verify the precision and reliability of the proposed method.
Comments on the Quality of English LanguageThe English could be improved to more clearly express the research.
